# Maintaining Golgi Homeostasis: A Balancing Act of Two Proteolytic Pathways

**DOI:** 10.3390/cells11050780

**Published:** 2022-02-23

**Authors:** Ron Benyair, Avital Eisenberg-Lerner, Yifat Merbl

**Affiliations:** Department of Immunology, Weizmann Institute of Science, Rehovot 7610001, Israel; rbenyair@umich.edu (R.B.); avital.eisenberg@mail.huji.ac.il (A.E.-L.)

**Keywords:** Golgi, proteostasis, EGAD, GARD, GOMED, autophagy, proteasomal degradation

## Abstract

The Golgi apparatus is a central hub for cellular protein trafficking and signaling. Golgi structure and function is tightly coupled and undergoes dynamic changes in health and disease. A crucial requirement for maintaining Golgi homeostasis is the ability of the Golgi to target aberrant, misfolded, or otherwise unwanted proteins to degradation. Recent studies have revealed that the Golgi apparatus may degrade such proteins through autophagy, retrograde trafficking to the ER for ER-associated degradation (ERAD), and locally, through Golgi apparatus-related degradation (GARD). Here, we review recent discoveries in these mechanisms, highlighting the role of the Golgi in maintaining cellular homeostasis.

## 1. Introduction

The Golgi, first described in 1898 by Camillo Golgi, is a stacked membranous organelle that serves as a hub of protein trafficking and post-translational modifications [1,2]. While traversing the Golgi, secretory glycoproteins undergo a series of modifications, wherein sugars are removed and added to their glycan chains, resulting in microheterogeneity of secreted glycoproteins [3]. Secretory proteins are sorted at the Golgi and are targeted to their final destinations, which include the plasma membrane, the endomembrane system, and secretion to the extracellular milieu. 

Secretory proteins undergo strict quality and quantity control processes that monitor their proper folding, complex assembly, post-translational modifications, and correct targeting to organelles [4,5,6,7]. The first quality control checkpoint that secretory proteins undergo occurs co-translationally, during their synthesis by endoplasmic reticulum (ER)-bound ribosomes. Properly folded proteins exit the ER via ER exit sites (ERES) towards the Golgi apparatus. Damaged or unfolded proteins, however, are targeted by quality control machinery to degradation by the two main protein degradation pathways, namely the ubiquitin proteasome system (UPS) and lysosomal degradation. Canonically, cellular proteasomes are considered the main degradation machinery for cytosolic proteins, while transmembrane or secreted proteins are thought to be targeted to lysosomal degradation. However, this distinction is more complex when considering proteins along the secretory pathway, wherein proteasomes facilitate a large part of ER-associated degradation (ERAD). During ERAD, proteins that fail to pass the quality control checkpoint at the ER are translocated across the ER membrane and are degraded by proteasomes in proximity of the ER (Reviewed in [8,9]). Yet, some proteins are degraded by alternative lysosomal degradation routes. Lysosomal degradation of secretory proteins may be mediated either by direct vesicular trafficking to lysosomes or via autophagy, a process in which double membrane vesicles are formed de novo around substrates that are recognized by the autophagic receptor protein p62, or even around parts of organelles such as the ER. Newly formed autophagosomes then fuse with cellular lysosomes leading to the degradation of the engulfed proteins/content by lysosomal proteases [10,11,12]. 

The multiplicity of potential degradation routes for secretory proteins raises various questions regarding the criteria for selection of the degradation mechanism. While it is clear that the mere distinction of cytosolic vs. membranal or luminal proteins is not sufficient to explain degradation in the secretory pathway, it is intriguing to decipher the determinants that dictate selectivity and specificity of substrate degradation. Is it protein dependent or context specific? What happens upon failure to degrade in the primary degradation mechanism? Why are some proteins degraded at the ER while other escape and are recognized at a post-ER checkpoint? Or namely, what determines the intracellular localization of degradation? Particularly, how is degradation facilitated at the post-ER sites, such as the Golgi apparatus? 

Here, we review the advancements made in the understanding of the Golgi apparatus and its role as a major subcellular hub of proteostasis, sensing and targeting proteins to degradation by the proteasomal and lysosomal degradation systems. We will describe the intricate complementary connection of the Golgi with the autophagic machinery and discuss recent findings demonstrating ERAD-independent proteasomal degradation at the Golgi. 

## 2. Proteasomal Degradation and the Golgi

ERAD is a major mechanism for quality and quantity control of proteins in the secretory pathway. ERAD facilitates the degradation of ER membrane or luminal proteins via proteasomes, which are recruited to the ER membranes [13]. Over the years, numerous lines of research have alluded to the question of what happens to damaged proteins that escape the ER, or how is quality control of post-ER processes that occur at the Golgi, such as different modifications, regulated. Many studies describe that while the Golgi can serve as a sensor of quality control, proteasome-dependent degradation requires an additional step of retrieval of proteins from the Golgi to the ER, and their subsequent degradation by proteasome-dependent ERAD. For example, studies in yeast demonstrated that ERAD degradation of proteins that are expressed in excess, such as mutant vacuolar carboxypeptidase Y (CPY*) and Proteinase A (PrA*), requires cycling between the ER and Golgi [14,15,16,17]. Interestingly and perhaps counter-intuitively, for some proteins, exit from the ER was proven as a pre-requisite for degradation by ERAD [18,19]. For example, unassembled MHC molecules are degraded by ERAD only after reaching the cis-Golgi and subsequent retrieval to the ER [20]. Such findings raise the notion that there is more than merely overflow of the ER in the ‘decision’ to exit the ER prior to proteasomal degradation. It is yet unclear why some proteins are targeted to ERAD in the ER, while others must first reach the Golgi. IgM degradation in B cells is an interesting example that demonstrates the complexity of degradation routes in the secretory system. Specifically, the route of IgM degradation in B cells is associated with the differentiation stage of the cells. In early differentiation stages, pre-B cells produce an excess of a soluble form of IgM, which is efficiently degraded [21]. The degradation of the soluble IgM μ heavy chain was shown to be proteasome-dependent and to occur at a post-ER, pre-trans-Golgi, compartment [21,22,23]. Here, too, as in the case of MHC, it appears that proteins are retrieved to the ER before reaching the trans-Golgi, suggesting a role for the cis-Golgi in sorting substrates for ERAD. In support, retrieval to the ER and ERAD of misfolded transmembrane proteins occurs through their recognition by either the K/HDEL retrieval receptor, Rer1 [24], or Erv29, a COPII vesicle cargo receptor [17,25], both residing at the cis-Golgi.

The main signal associated with targeting of proteins to proteasomal degradation is ubiquitination by E3 ubiquitin ligases. Various ubiquitin E3 ligases, as well as deubiquitinating enzymes (DUBs) are known to be recruited to the Golgi apparatus and regulate the degradation of proteins. Yet, for many years, Golgi-localized ubiquitination was mainly reported in the context of trafficking or lysosomal degradation [26,27]. Nevertheless, several studies described phenomena in which proteins at the Golgi were targeted for proteasomal degradation that did not involve the ER or lysosomes. For example, in fission yeast, the sterol regulatory element-binding protein (SREBP) is proteolytically cleaved by Rhomboid 2 and Cdc48, following ubiquitination by the E3 ligase Dsc, which is localized to the Golgi apparatus [28,29]. The C-terminal fragment of SREBP is transported back to the ER following Rhomboid cleavage and is degraded by ERAD [30] while the N-terminus of the cleaved SREBP is translocated to the nucleus to act as a transcription factor [28]. Interestingly, in the absence of Rhomboid cleavage, the SREBP precursor is targeted to proteasomal degradation in a manner that is dependent on Dsc E3 ligase activity, and independent of ERAD E3 ligases Hrd1 and Doa10 [28]. The route that the SREBP precursor undertakes from the Golgi membrane to the proteasome was, however, not described.

The identification of endosome and Golgi-associated degradation (EGAD) in budding yeast [31] sheds new light on the role of proteasomes in Golgi-associated degradation (Figure 1). In EGAD, Golgi membrane proteins undergo proteasomal degradation without retrieval to the ER, but rather through their targeting from the Golgi to cytosolic proteasomes [31]. Interestingly, the first demonstration of EGAD involved the E3 ligase Dsc, which, as mentioned above, was likewise reported in fission yeast to mediate the ERAD-independent proteasomal degradation of SREBP by an uncharacterized mechanism [28]. While SREBP homologs are absent from budding yeast, the Golgi-localized Dsc E3 ligase complex was demonstrated in this strain to induce the polyubiquitination of the Golgi membrane protein Orm2 [31]. Yet, in contrast to most cases wherein polyubiquitinated proteins at the Golgi are sorted by ESCRT components to vacuolar/lysosomal degradation, Orm2 is extracted from the membrane following its ubiquitination via the function of the ATPase VCP/CDC48 and is targeted for degradation by cytosolic proteasomes [31]. As Orm2 is a negative regulator of sphingolipid biosynthesis, the post-ER checkpoint of EGAD is key to regulate lipid metabolism in budding yeast. It is plausible that Orm2 degradation may be the consequence of sphingolipid-sensing quality control mechanisms at the Golgi. Additional EGAD substrates and a potentially homologous mechanism in mammalian cells remain to be fully elucidated.

The identification of EGAD demonstrated that proteins may be targeted to proteasomal degradation independently of ERAD. Yet, in contrast to ERAD, in the case of EGAD, proteasomes were not shown to be associated with the Golgi or endosomes, but were mainly localized to vesicles in the cytosol. A possible explanation may be the fact that the Golgi apparatus in yeast has a simplified, unstacked, architecture and is localized randomly in proximity to the ER. How then is ERAD-independent proteasomal degradation of Golgi-localized proteins regulated in mammalian cells? Recent findings demonstrated a novel mechanism of Golgi apparatus-related degradation (GARD) (Figure 1) that involves proteasomal degradation via proteasomes that are associated with Golgi membranes [32]. Specifically, proteasomes at the Golgi compartment were shown to be activated in response to Golgi-stress stimuli, such as block of Golgi trafficking or inhibition of sialylation, leading to the activation of GARD [32]. Such stress-induced activation of GARD was shown to be critical for regulating Golgi morphology, which is maintained by a series of structural proteins such as GM130 [32,33]. Golgi-stress induced the Golgi-localized degradation of the Golgi tethering factor GM130 through GARD, leading to Golgi dispersal. The block of proteasome activity, on the other hand, prevented Golgi dispersal under Golgi stress. Interestingly, these findings support and perhaps relate to previous work in neurons that described the proteasomal degradation of another Golgi tethering factor, GRASP65, following its ubiquitination by the Cul7-FbxW8 E3 ligase complex [34]. In that case too, the regulation of GRASP65 turnover controlled Golgi morphology [34]. Whether GRASP65 is targeted to degradation via Golgi-localized proteasomes in a Cul7-FbxW8-dependent manner remains to be determined.

Interestingly, Golgi dispersal via GARD is reversible, allowing the Golgi ribbon to recover its distinctive morphology when the stress is removed. In contrast, extended stress leads to irreversible changes and induces cell death in a manner that is dependent on GM130 degradation. Thus, localized proteasomal degradation allows for a rapid response to Golgi stress, providing a temporal window of response to control cell fate by either regaining Golgi integrity or inducing cell death. GARD-dependent regulation of Golgi stress was shown to be key in the highly secretory malignancy, multiple myeloma. Activation of GARD under Golgi-stress conditions led to cell death in multiple myeloma cells [32]. It is plausible to assume that multiple myeloma cells are particularly sensitive to changes in the Golgi, such as those induced by GARD, due to their extended secretory system, and their high dependence on intact Golgi structure and function. The Golgi stress-induced mortality of multiple myeloma cells proved beneficial in a mouse model of multiple myeloma, wherein treatment with the Golgi-stress inducer monensin dramatically reduced the number of malignant cells in the mouse [32]. While this work demonstrated the role of GARD in controlling cell death via regulation of Golgi morphology, the full range of GARD substrates and the effect of their degradation remains to be elucidated. Further, whether GARD serves to control only Golgi morphology, or may also provide a mechanism of quality control for proteins traversing through the Golgi remains to be investigated.

### Quality Control and Stress Response at the Golgi

Protein quality control (PQC) mechanisms exist in numerous cellular compartments, beyond the ER. Ribosomes, mitochondria, the plasma membrane [5,35,36,37], and the Golgi [4] have all been reported to be involved in cellular homeostasis through localized PQC mechanisms. Such mechanisms provide continuous control over the fidelity of protein function, downstream of ERAD. For example, in yeast, the expression of a mutated, unstable form of bovine pancreatic trypsin inhibitor (BPTI) protein caused the accumulation of this secretory protein in the Golgi, from which mutant BPTI was targeted for vacuolar degradation [38]. Another secretory protein in yeast, Wsc1p, is able to evade ER quality control mechanisms despite mutations that cause misfolding of its luminal domain. Mutated Wsc1p is targeted to vacuolar degradation via the Golgi, suggesting that this protein undergoes its major quality control process in the Golgi, as opposed to the ER [39]. In mammalian cells, Briant et al. introduced different transmembrane domains into the secretory co-receptor CD8. While two of these domains caused retrieval of CD8 from the Golgi to the ER, a third domain targeted CD8 to lysosomal degradation directly from the Golgi [40]. These findings indicate that the Golgi quality control machinery can differentially target proteins to retrograde trafficking vs. degradation based on their transmembrane domains. In a recent study, Hellerschmied et al. used the Golgi-targeting sequences of MAN2A1 and either B4GALT1 or ST6GAL1, to target EGFP proteins to either cis- or trans-Golgi, respectively. These proteins also expressed a HaloTag2 domain, which can be chemically induced to unfold and expose hydrophobic domains [41]. Using this system, the authors could target Golgi-localized EGFP to lysosomal degradation and showed that unfolded proteins are identified and segregated from folded proteins within the Golgi, a critical ability in protein quality control, reminiscent of the ability of the ER to segregate and compartmentalize ERAD substrates [42].

## 3. Autophagy and the Golgi Apparatus

The Golgi apparatus and the autophagic machinery influence each other in complementary ways. On the one hand, the dynamics of membrane organization and protein trafficking through the Golgi control the induction of autophagy and elongation of autophagosomes, while on the other hand, autophagy can affect the Golgi structure and function (Figure 2). In this section, we discuss both these processes and their importance in maintaining the integrity of the secretory system and cellular function.

The autophagic machinery: autophagy is initiated by cell stress, such as shortage in amino acid or sugar availability. During autophagy, ubiquitinated proteins are identified by autophagy receptors such as p62/SQSTM1 which, in turn, associate with autophagy-related 8 (Atg8) family proteins such as microtubule-associated protein 1 light chain-3 (LC3) and/or GABARAP. During initiation of autophagy, an enzymatic cascade initiated by the activity of ULK1 and the Beclin-PI3KIII complex, induces the nucleation of the nascent autophagosome, followed by the recruitment of additional components, such as LC3, that contribute to phagophore elongation. LC3 is then modified by the addition of a phosphatidylethanolamine (PE) lipid, which facilitates the binding of LC3 to the expanding autophagosome membrane. Lipidated LC3 also binds p62/SQSTM1, leading to the recruitment of p62 along with its ubiquitinated protein targets to the autophagosome. In the process of maturation, the autophagosome will seal off in a double-membrane structure that then fuses with the lysosome, creating the autolysosome in which autophagic cargo is hydrolyzed, allowing the recycling of amino acids and sugars (Reviewed in [54]).

### Maintenance of Golgi Morphology and Trafficking Are Linked to Autophagy

Various studies have established that autophagy is regulated by Golgi dynamics by demonstrating the impact of the critical proteins that control Golgi morphology, such as golgins and SNARE proteins, on autophagosome formation and maturation. These studies, described below, suggest that autophagy and Golgi structure are intertwined. Under certain conditions, proper Golgi structure is required for local initiation of autophagy, while under different conditions, Golgi fragmentation is conducive to autophagy via freeing up of Golgi-associated autophagy factors.

The secretory pathway is essential for the initiation of autophagy and formation of autophagosomes [55,56] (Figure 2). For example, AMDE-1, a chemical inducer of non-canonical (ulk1/Beclin-independent) autophagy, induced the accumulation of lipidated LC3 at the Golgi [57]. AMDE-1-induced LC3 lipidation was inhibited by Golgi dispersal and by V-ATPase inhibitors, suggesting that the Golgi could function as a platform for LC3 lipidation that is dependent on V-ATPase [58]. Interestingly, under conditions that disturb trafficking to the plasma membrane, overwhelming the Golgi with substrates such that the Golgi expands, was shown to induce the accumulation of LC3 on trans-Golgi network (TGN) membranes [56].

The Golgi reassembly stacking proteins 55 (GRASP55) and 65 (GRASP65) were characterized in the late 1990s [59,60] and were hypothesized to be crucial for maintenance of Golgi stacking [61]. More recent work has shown that GRASP55 and GRASP65 are in fact dispensable for the maintenance of Golgi morphology, as their acute degradation or downregulation by RNAi does not induce Golgi fragmentation, and mice lacking these displayed properly stacked Golgi. However, long-term concomitant depletion of both GRASPs was shown to cause Golgi fragmentation, most likely due to reduced stability of the GRASP65 interacting protein, GM130 [62,63]. A role for GRASP55 was suggested in the initiation of autophagy. Knockdown of GRASP55 caused a reduction in LC3 lipidation, as well as a reduction in autophagosome abundance, but not autolysosome abundance, suggesting that GRASP55 is involved in the initiation, but not maturation, of autophagosomes [64]. Conversely, other work has shown that GRASP55 knockdown caused an increase in LC3 lipidation and autophagosome formation, yet reduced the amount of autolysosomes together with an increase in p62 abundance. These results suggested that GRASP55 has an inhibitory role in the initiation of autophagy, but is important in autophagic maturation [65]. The authors suggested a mechanism by which GRASP55 is normally O-GlcNAcylated under steady state conditions, but loses this modification under starvation conditions. This in turn causes GRASP55 to change its localization from the Golgi to autophagosomes and late endosomes/lysosomes, where it directly interacts with LC3 and the lysosomal LAMP2. Interestingly, GRASP55 has also recently been found to be a substrate of mTORC1, a major sensor and initiator of autophagy. The inhibition of mTORC1 induced the de-phosphorylation of GRASP55 and its re-localization to multivesicular bodies (MVBs), where GRASP55 likely plays a role in exosome secretion [66].

The GRASP65 interacting protein, GM130, is a cis-Golgi-localized tethering protein, important in maintenance of Golgi morphology [67,68]. At steady state, GM130 acts as an inhibitor of autophagy by binding GABARAP, keeping it sequestered at the Golgi membrane. Under starvation conditions, GM130 was shown to bind WW domain-containing adaptor with coiled coil (WAC) and to facilitate its association with the Golgi. The competitive binding of WAC to GM130 inhibits GM130 association with GABARAP, releasing it to fulfill its role in phagophore formation [51,52,53]. Another golgin that acts as a negative regulator of autophagy is GCC88, a TGN-localized protein that facilitates the formation of a complete Golgi ribbon from Golgi mini-stacks [69]. Aberration of the Golgi ribbon morphology by knockdown of GCC88 lead to the inhibition of mTOR, an increase in autophagosome abundance, and LC3 lipidation, indicating the initiation of autophagy [70].

SNARE complex proteins play a crucial role in membrane fusion [71]. At the Golgi, SNAREs are involved in the docking of ER-derived vesicles as well as in intra-Golgi trafficking [72]. Some SNARE complex components are associated with Golgi morphology as well as autophagy. The soluble N-ethylmaleimide-sensitive factor attachment protein α (αSNAP) is important in the disassembly of SNARE complexes, following membrane fusion, as well as in SNARE assembly [73]. Depletion of αSNAP was shown to cause Golgi fragmentation and concomitantly increases LC3 lipidation, causing an accumulation of autophagosomes that contain Golgi markers. This initiated non-canonical autophagy, which was dependent on Atg5/Atg7 pathway, but independent of Beclin1 and Vps34 [74]. Another Golgi-localized SNARE protein is syntaxin-5, that has been shown in yeast to take part in ER to Golgi trafficking [75]. Knockdown of syntaxin-5 was shown to cause Golgi fragmentation as well as an accumulation of LC3-positive puncta, and impaired clearance of autophagy substrates [76,77]. Syntaxin-17 is a SNARE protein that cycles between the ER and the ER to Golgi intermediate compartment (ERGIC). Syntaxin-17 knockdown resulted in dispersal of the ERGIC structure and in fragmentation of the Golgi [78]. During the initiation of autophagy, syntaxin-17 is phosphorylated at S202 by TBK1. This in turn causes a shift in the localization of syntaxin-17 to the Golgi and from there to the cytosolic Ulk1 complex which initiates phagophore formation [79].

The Golgi-localized scaffold protein PAQR3, important in vesicle fission and ER to Golgi COPII vesicle trafficking, is a positive regulator of autophagy [73,80,81,82]. Overexpression of PAQR3 causes Golgi fragmentation by regulating plasma membrane-bound trafficking [81]. PAQR3 is proposed to be an important initiator of autophagy, by acting as an inhibitor of mTORC1 activity via the reduction in the interaction of mTORC1 with raptor and mLST8 [83]. PAQR3 also associates with Atg14L and Vps34 to facilitate the initiation of autophagy by the Atg14L-Vps34 complex. PAQR3 is phosphorylated by AMPK upon glucose starvation, and activates the production of PI3P by Vps34, initiating autophagosome formation [73].

CLEC16A regulates autophagy through its association and modulation of mTOR [80]. CLEC16A was shown to localize both to the Golgi and to cytosolic vesicles, while starvation conditions cause a shift of this protein to a mainly Golgi localization. Interestingly, mutations in CLEC16A which cause CLEC16A deficiency also caused abnormalities in autophagy and dispersal of the Golgi apparatus [84]. Golgi phosphoprotein 3 (GOLPH3) localizes to the Golgi through its binding of PI4P and links the Golgi to the actin cytoskeleton by binding the myosin MYO18a and is also involved in vesicle trafficking by acting as a COP-I vesicle adaptor [85,86]. GOPLH3 has been implicated in Golgi stress response. In a model of glucose deprivation and hypoxia, followed by reoxygenation, GOLPH3 expression was shown to increase along with LC3 lipidation over time. Under the same conditions, but with the addition of GOLPH3 knockdown, LC3 lipidation was not increased, suggesting that GOLPH3 positively regulates autophagy initiation [87]. More recently, Lu et al. showed that GOLPH3 directly interacts with LC3 in the initiation of Golgiphagy, suggesting a role for GOLPH3 in Golgi homeostasis. However, knockdown of GOLPH3 inhibited the autophagic targeting of Golgi proteins under conditions of Golgi stress [88]; thus, the role of GOLPH3 in Golgiphagy remains to be determined. Under conditions of DNA damage, GOLPH3 is phosphorylated, causing fragmentation of the Golgi, and knockdown of GOLPH3 prevents this fragmentation under these conditions [89]. In another study however, the knockdown of GOLPH3 was shown to inhibit the activity of mTOR under EGF treatment while GOLPH3 overexpression caused an increase in mTOR activity, suggesting GOLPH3 could be an inhibitor of autophagy [90].

In yeast, TOR inhibition by starvation was shown to cause ubiquitination of Golgi quality control substrates by the Dsc complex, targeting substrates to the MVB-to-lysosome pathway and ultimately to lysosomal degradation [91]. One such substrate is Tlg1, a SNARE protein associated with intra-Golgi trafficking in yeast [92]. Tlg1 receives a palmitoyl modification by Swf1, facilitating tlg1 localization at the Golgi. In the absence of palmitoylation, Tlg1 stability and association to the Golgi membrane are compromised. Tlg1 is then ubiquitinated by Tul1, a Golgi-localized ubiquitin E3 ligase identified in yeast and targeted to lysosomal degradation through the MVB pathway [27,93].

Rab proteins act as GTPases that modulate SNARE protein activity [94] and are primarily involved in both vesicle trafficking [95,96] and autophagy [97]. There are approximately 60 known Rab protein variants, a third of which are Golgi associated and play a role in Golgi morphology [98,99]. Several of the Golgi-associated Rab GTPases have been implicated in autophagy [100,101]. Rab1 and Rab2 are involved in trafficking between the ER and Golgi [102]. Rab1 was shown to associate with Golgin-84, GM130 and p115 [103]. Mutations in Rab1 or knockdown of Rab1 expression inhibited autophagy initiation, formation of LC3 puncta and LC3 lipidation [104]. It is unclear however, whether this is due to aberrations of Rab1 activity at the ER or the Golgi. In yeast, Rab2 was shown to associate with Rab7 as part of a HOPS-dependent pathway for lysosomal degradation of autophagosomes and endosomes [105]. In mammalian cells, Rab2 keeps its Golgi localization via interaction with GM130. Upon starvation, Rab2 leaves the Golgi and interacts with the Ulk1 complex, promoting phagophore formation, leading to autophagosome and autolysosome formation through modulation of Ulk1 phosphorylation [106].

Rab6 is a Golgi-localized Rab protein that is required for intra-Golgi trafficking [107]. In *Drosophila* and yeast, Rab6 (or its yeast ortholog Ypt6) was shown to be important in the trafficking of lysosomal/vacuolar components. Knockdown of Rab6 in *Drosophila* lead to the enlargement of autolysosomes and reduced degradative capacity [108,109].

Rab9 is a late endosomal Rab protein that functions in endosome to Golgi vesicular transport [110]. Rab9 is primarily required for Atg5/7-independent autophagy [111,112] but is also required for the recycling of mannose 6-phosphate (M6P) receptors from late endosomes back to the Golgi. M6P is specifically added to the N-linked glycan structures of lysosomal proteins as they traverse the Golgi. M6P-tagged proteins are then identified at the Golgi by M6P receptors, which target these proteins for trafficking to the lysosome. M6P receptors are then recycled back to the Golgi by vesicular trafficking that requires Rab9. Aberrations in the trafficking of lysosomal proteins from the Golgi can cause the depletion of lysosomal enzymes, the inhibition of autophagy and lysosomal storage disorders, making Rab9 important in lysosomal function [113,114].

Rab33 localizes to the Golgi and is involved in intra-Golgi trafficking. Rab33B was shown to interact with Atg16L and overexpression of Rab33 even recruited Atg16L to the Golgi, precluding it from acting in autophagosome maturation. Atg16L acts as a scaffolding protein that facilitates LC3 lipidation and membrane association in maturing phagophores [115]. The interaction between Rab33 and Atg16L at the Golgi under non-starvation conditions was shown to inhibit autophagy but interestingly, knockdown of Rab33 did not have an effect on autophagy [116]. More recently, Rab33B was shown to translocate from the Golgi to phagophores under starvation conditions, and recruit Atg16L to fulfill its function in autophagy, making Rab33 an important player in the initiation of autophagy [117].

Atg9 (Atg9A in mammalian, Atg9 in yeast) is a large Golgi-localized transmembrane protein that is essential for the initiation of autophagy and autophagosome formation [118,119]. Knockout of Atg9 results in reduced LC3 lipidation and puncta formation in response to starvation, as well as an accumulation of the autophagy adaptor and substrate p62 [120]. Atg9 constitutively cycles between the Golgi and late endosomal compartments in steady-state conditions [43,44]. Under starvation, Atg9 associates with the clathrin adaptor AP1 subunit, γ-adaptin, which promotes the redistribution of Atg9 to autophagosomes [49]. This mechanism was shown to be mediated via the phosphorylation of Atg9 at Tyr8 which induces the binding of Atg9 to the AP1 (as well as AP2 and AP4) complex [49], and the phosphorylation of Ser14 on Atg9 by ULK1, which facilitates a shift in localization of Atg9 from the Golgi to a peripheral, LC3-positive, autophagosomal compartment, also known as the Atg9 compartment [43,46,47,49,121,122]. This step is further regulated via the recruitment of Ulk1 itself to the Golgi, through its phosphorylation at Ser-746 by RIPK3. When phosphorylated at Ser-746, Ulk1^746^ localizes to the Golgi and is required for alternative, but not canonical, autophagy [45]. The redistribution of Atg9 is a key step in the initiation of autophagy and autophagosome formation, making the Golgi apparatus an important player in these processes [46,47,48,49].

Interdependency between Golgi morphology and Atg9 trafficking was shown to regulate autophagy. The mammalian transport protein particle complex (TRAPP) plays a role in COPII vesicle formation [123], and is involved in the cycling of Atg9 from recycling endosomes back to the Golgi via the TRAPPC8 subunit [124]. Depletion of TRAPPC8 induced Golgi fragmentation and inhibited autophagosome formation [125]. Another TRAPP subunit, TRAPPC13, is involved in the action of Rab1a and Rab1b which play roles in ER to Golgi trafficking [126]. Knockdown of TRAPPC13 inhibited Rab1 activity and protected the Golgi from multiple disrupting agents. TRAPPC13 knockdown also inhibited the lipidation of LC3, as well as the formation of LC3-positive puncta under Brefeldin A1 (BFA) treatment [127], strengthening the link between this complex, which is important in the maintenance of Golgi morphology and autophagy.

Additional examples demonstrate the tight link between proteins required for maintaining Golgi morphology to Atg9 trafficking and regulation of autophagy. The conserved oligomeric Golgi (COG) complex is an octoheteromeric tethering complex involved in membrane trafficking and is crucial for the maintenance of Golgi morphology and function in both mammalian and yeast cells [128,129]. In yeast, the COG complex has been shown to directly regulate Atg9 trafficking and selective autophagy. Mutations in COG-related genes caused the mis-localization of Atg9 and inhibition of autophagy [83,130].

p230, a TGN-localized golgin protein positively regulates autophagy. In p230 knockdown cells, starvation failed to increase LC3 lipidation and autophagic flux and caused reduced Atg9 recruitment from the TGN to peripheral autophagosome-associated membranes, suggesting a role for the Golgi protein p230 in the initiation of autophagy via Atg9 [131].

Syntaxin-16 is a Golgi-localized SNARE [132,133] that has recently been suggested to be involved in autophagosome formation and autolysosome biogenesis by facilitating Atg9 trafficking through interactions with VAMP7 [134,135].

BAR-domain proteins induce membrane curvature and recruit cytosolic proteins that support membrane trafficking [136]. Vesicular trafficking of Atg9 is also modulated by BAR-domain proteins. The BAR-domain protein Bif1 interacts with the autophagy-associated Vps34 complex-II through UVRAG. Knockdown or mutation of Bif1, or knockdown of UVRAG/Beclin1 all inhibited the change in localization of Atg9 from the Golgi to the peripheral compartment, causing an inhibition in autophagy [137]. The BAR-domain protein SNX18 promotes autophagy through membrane remodeling [138]. SNX18 regulates trafficking of Atg9 from the Rab11 positive recycling endosomes, another source of Atg9 that is important in autophagy, to autophagosome membranes [47,139]. Arfaptin-2, a BAR-domain protein, was detected by mass-spectrometry analysis in Atg9-positive vesicles, which also contain Bif1, under amino acid starvation conditions [50]. These vesicles also contained the PI4P kinases PI4KIIα PI4KIIIβ. PI4KIIα and PI4KIIβ are TGN-localized enzymes that are recruited to autophagosomes by GABARAP to produce the phosphoinositide lipid PI4P, which in turn promotes autophagosome–lysosome fusion [140]. PI4P is abundant in the Golgi and is important in the maintenance of Golgi morphology and vesicular trafficking [141,142], as well as in autophagy [50,140]. This suggests that Atg9 may play a role in providing phosphoinositide-metabolizing enzymes to autophagosomes, allowing the local production of PI4P, crucial for autophagy. The levels of PI4P at the Golgi are further regulated by the PI4P phosphatase SAC1. SAC1 is a type-II transmembrane protein localized to the ER, and is trafficked from the ER to the Golgi under starvation conditions, where it increases autophagosome formation [143]. In addition to canonical autophagy, an Atg5/Atg7 independent autophagy pathway was recently described, by which proinsulin granules are targeted to lysosomal degradation from the Golgi. In both yeast (which lack Atg5) and in mammalian Atg5/Atg7 knockout cells, this Golgi membrane-associated degradation (GOMED) pathway was activated by the disruption of PI4P trafficking to the plasma membrane [144].

Despite the knowledge accumulated about Atg9, its localization in steady state and during starvation, and its various interactors, the precise role of Atg9 remains to be elucidated. Whether Atg9 solely serves to provide membranes from the Golgi to autophagosomes [44,145] or has additional roles such as the delivery of metabolizing enzymes [146], it is clear that this Golgi-localized protein is required for the ramp-up of autophagosome formation early in starvation-induced autophagy [121].

## 4. Discussion and Outlook

The existence of diverse routes for degradation at the Golgi, including lysosomal degradation, retrieval for ERAD, and Golgi-associated proteasomal degradation by EGAD and GARD, exemplify the complexity of quality control mechanisms in the secretory pathway and raise intriguing questions. First, why are several mechanisms of proteasome-dependent degradation at the Golgi, involving either retrieval to the ER or localized degradation, required? Second, are there specific protein determinants involved in directing substrates to these different paths? Further, what are the sensing mechanisms involved in protein-fate decisions downstream of the ER? An example suggesting that specific protein determinants may direct the route of degradation was given by the substitution of the type I transmembrane domain (TMD) of CD8 with a 4-pass TMD and subsequent deletions within the TMD [40]. Briant et al. identified that aberrations in different regions in the TMD led to distinct degradation mechanisms. While some mutant forms were retained in the ER and degraded by ERAD, others, such as a mutant lacking three polar residues required for Rer1 recognition, exited the ER and reached the Golgi. However, this form was unstable following the escape from the ER and was degraded in a proteasome-dependent manner. Although the mechanism by which this was mediated was not examined, the authors hypothesized that some of the protein does undergo ERAD before transport to the Golgi [40]. It would, nevertheless, be interesting to examine whether localized proteasomal function at the Golgi may be involved in the degradation of the mutant TMD-fused protein as well. Thus, full characterization of the features that define the route to proteasomal degradation at the Golgi and other components of the secretory pathway is still required. Research showing that misfolded proteins can be identified and sequestered in the Golgi raises intriguing questions of whether the Golgi can also act as a quality control hub, identifying and targeting misfolded or incorrectly modified proteins to degradation via autophagy or proteasomes. Future studies may reveal the role of Golgi-associated proteasomal degradation in quality control mechanisms and how these are differentiated from ones of ERAD. 

Harnessing Golgi-associated degradation for translational purposes is a vital outlook. For example, aberrations in sialylation, a form of glycosylation that occurs at the Golgi, are implicated in cancer and were shown to be positively associated with metastasis, cell survival and tumor progression [147,148]. Disrupting Golgi function via Brefeldin A, a natural antiviral compound that causes the redistribution of the Golgi to the ER, caused tumor cell death and the inhibition of proliferation in melanoma and prostate carcinoma cells [149]. Furthermore, inducing degradation-dependent dispersal of the Golgi via treatment with monensin, a Golgi-pH neutralizing ionophore, or lithocholylglycine (LCG), an inhibitor of sialylation, were shown to induce cell death of multiple myeloma cells in a manner that was dependent on Golgi-associated degradation of the Golgi structural protein GM130 [32]. Moreover, treatment with monensin was successful in alleviating the progression of multiple myeloma in an in vivo mouse model, by reducing the amount of circulating multiple myeloma cells and inhibiting splenomegaly, a hallmark of multiple myeloma. Thus, it is intriguing to speculate that aberrations in protein structure, post-translational modification, or quantity may be sensed at the Golgi and thereafter induce the localized proteasomal degradation of the damaged substrates. Whether that is the case remains to be determined.

Beyond quality control, a key feature of both autophagy and proteasomal degradation is the modulation of Golgi morphology. Regulation of Golgi dynamics and integrity is crucial in specialized secretory cells, such as hepatocytes, antibody-secreting plasma cells, neurons and others, wherein aberrations in the secretory flow of proteins may overwhelm the secretory organelles leading to cytotoxic damage and cell death. In fact, in several neurodegenerative diseases, Golgi fragmentation has been observed in neurons as an early event in neurodegeneration, preceding other pathological phenotypes [150,151]. It is therefore important to realize how control of degradation events at the Golgi impacts the homeostatic balance of secretory cells, and how is degradation regulated under stress such as increased protein synthesis or disruption of trafficking. Further studies are required to decipher the key players and mechanisms that are involved in sensing stress at the Golgi and eliciting changes in degradation. Furthermore, it remains to be determined if and how does crosstalk between the different degradative pathways control the functional fidelity of the secretory pathway.

## Figures and Tables

**Figure 1 cells-11-00780-f001:**
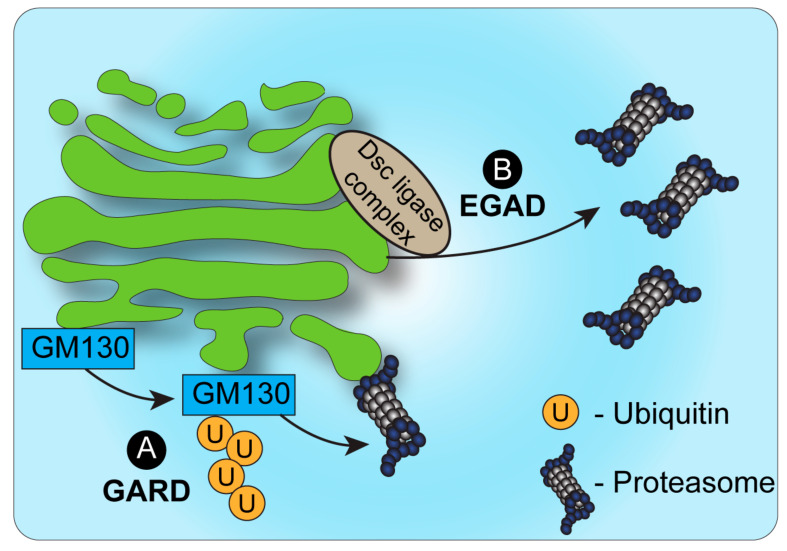
Proteasomal degradation and the Golgi. (**A**). Under conditions of Golgi stress, the structural Golgin protein GM130 is ubiquitinated and targeted for degradation by proteasomes, bound to the cytosolic side of the Golgi membrane. This process, known as Golgi apparatus-related degradation (GARD) allows the Golgi to regulate its morphology quickly in response to stress [32]. (**B**). In yeast, endosome and Golgi-associated degradation (EGAD) has been described as a mechanism by which proteins can be ubiquitinated by the Dsc complex, released from the Golgi membrane by VCP/CDC48 and degraded by cytosolic proteasomes [31].

**Figure 2 cells-11-00780-f002:**
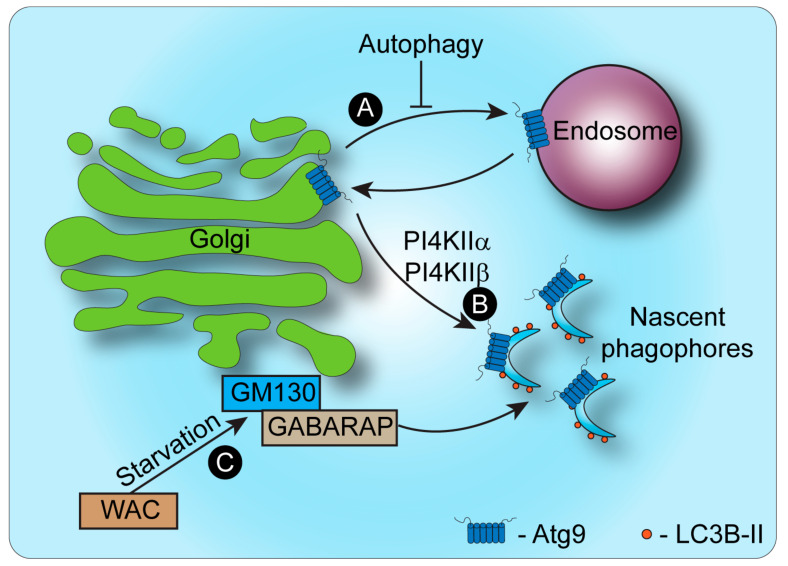
The Golgi apparatus in autophagy. (**A**). Atg9 actively cycles between the Golgi apparatus and endosomes [43,44]. Upon initiation of autophagy, Golgi-to-endosome trafficking is inhibited, causing Atg9 to preferentially localize to the Golgi [45]. (**B**). Atg9 is then trafficked to nascent phagophore structures to support initiation of autophagy, in vesicles that contain PI4KIIα and PI4KIIβ [46,47,48,49,50]. This step provides both membranes and phosphoinositide lipid modifying enzymes crucial for initiation of autophagy. (**C**). At steady state, the autophagy-associated protein GABARAP, binds GM130 at the Golgi apparatus. Under starvation conditions, competitive binding of WAC to GM130 releases GABARAP to perform its role in initiation of autophagy [51,52,53].

## Data Availability

Not applicable.

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
