# Peer review of "Maintaining Golgi Homeostasis: A Balancing Act of Two Proteolytic Pathways"

_cells, 2022, doi:10.3390/cells11050780_

Round 1

Reviewer 1 Report

Review of Benyair et al

            The authors have put together a spectacular and informative review on the topic of protein degradation pathways and quality control at the level of the Golgi apparatus. This review is timely and topical. This review is very readable, clear and yet authoritative and detailed. In reading this manuscript I learnt important new details about protein handling within and related to the Golgi, and its interfacing with autophagosomes and proteasomes. The two figures are simple and very understandable. I enthusiastically support this manuscript.

Minor linguistic, typo and formatting issues on the following lines:

Fig. 1 and Fig. 2 are misnumbered and out of sequence with respect to the relevant flow in the text.

148, 149, 150, should be

Yet, in contrast to ERAD, in case of EGAD, proteasomes were not shown to be associated with the Golgi or endosome compartments, but were mainly localized to dots in the cytosol.

175, should be

Multiple myeloma cells, which are enriched in an extended secretory system, were shown to be particularly sensitive to cell death that was

182, should be

Interestingly, these findings support and perhaps relate to previous work in neurons that described the proteasomal degradation

format issue 236: Number 236 overlaps “The” in text box

502 should be

 the escape from the ER, and was degraded in a proteasome-dependent manner. Although

Author Response

We would like to thank the reviewer for their kind words and attentive review of our manuscript. We  corrected the text according to the  reviewer’s comments.

Reviewer 2 Report

The manuscript by Benyair, Eisenberg-Lerner and Merbl discusses the maintenance of Golgi homeostasis and the autophagy, ERAD and GARD mechanisms. The paper has an interest for broad scientific auditorium and provides interesting information. Most of the background and the cited literature is up-to-date and the structure is well designed. Overall, I consider the manuscript deals with an issue of interest and fits the scope of the journal. However, there are important limitations and a revision is required before publishing.

  • Please consider a major revision of the English style and grammar. It has a very poor style. Sometimes, it looks like it was reluctantly written:

(Line 8) The Golgi apparatus plays is a central hub for cellular protein trafficking and signaling.

(Line 20) While traversing the Golgi, secretory glycoproteins undergo a series of modifications, removing and adding sugars to their glycan chains, resulting in microheterogeneity of secreted glycoproteins.

(Line 85) Further provoking the question of why some proteins are targeted to ERAD in the ER, while others must first reach the Golgi, is the example of IgM degradation in B cells, wherein the route of degradation is associated with the differentiation stage of the cells.

(Line 169) Similarly, to ERAD, GARD is also increased under stress conditions of the Golgi.

(Line 174) Multiple myeloma cells, which are enriched with an extended secretory system, were shown to be particularly sensitive cell death.

(Line 184) Interestingly, these findings support and perhaps related to previous work in neurons that described the proteasomal degradation of another Golgi tethering factor, GRASP65, following its ubiquitination by the Cul7-FbxW8 E3 ligase complex (Litterman, Ikeuchi et al. 2011).

and there are misspelling such as (Line 149) Associat4ed.

  • Please, stick to abbreviations along the text. Explain abbreviations before using them (for example QC in line 172 is Quality Control?) and use them every time you use the word they refer to.
  • Figure 2 comes before Figure 1.
  • For section 3.1, a figure or table summarizing all the molecules explained along the text would help to understand their role and relationship.
  • Are there any differences between cell types for the mechanism described in the text? It would enrich the document if you could comment on it.

Author Response

We would like to thank the reviewer for their attentive review of our manuscript and suggestions for improvement. All errors brought forward by the reviewer were corrected and revised the style and grammar where applicable.